# Generalized Focal Loss: Learning Qualified and Distributed Bounding Boxes for Dense Object Detection

Xiang Li[1,2], Wenhai Wang[3,2], Lijun Wu[4], Shuo Chen[5,1], Xiaolin Hu[6], Jun Li[1], Jinhui Tang[1], and Jian Yang[1*]

[1]Nanjing University of Science and Technology [2]Momenta [3]Nanjing University
[4] Microsoft Research [5]RIKEN Center for Advanced Intelligence Project [6]Tsinghua University

{xiang.li.implus, jinhuitang, csjyang}@njust.edu.cn, wangwenhai362@163.com, shuo.chen.ya@riken.jp
lijuwu@microsoft.com, xlhu@mail.tsinghua.edu.cn, junl.mldl@gmail.com

## Abstract

One-stage detector basically formulates object detection as dense classification and localization (i.e., bounding box regression). The classification is usually optimized by Focal Loss and the box location is commonly learned under Dirac delta distribution. A recent trend for one-stage detectors is to introduce an *individual* prediction branch to estimate the quality of localization, where the predicted quality facilitates the classification to improve detection performance. This paper delves into the *representations* of the above three fundamental elements: quality estimation, classification and localization. Two problems are discovered in existing practices, including (1) the inconsistent usage of the quality estimation and classification between training and inference, and (2) the inflexible Dirac delta distribution for localization. To address the problems, we design new representations for these elements. Specifically, we merge the quality estimation into the class prediction vector to form a joint representation, and use a vector to represent arbitrary distribution of box locations. The improved representations eliminate the inconsistency risk and accurately depict the flexible distribution in real data, but contain *continuous* labels, which is beyond the scope of Focal Loss. We then propose Generalized Focal Loss (GFL) that generalizes Focal Loss from its discrete form to the *continuous* version for successful optimization. On COCO `test-dev`, GFL achieves 45.0% AP using ResNet-101 backbone, surpassing state-of-the-art SAPD (43.5%) and ATSS (43.6%) with higher or comparable inference speed.

## 1 Introduction

Recently, dense detectors have gradually led the trend of object detection. Based on dense detectors, researchers focus more on the *representation* of bounding boxes and their localization quality estimation, leading to an encouraging advancement [26, 29] in the field. Specifically, bounding box *representation* is modeled as a simple Dirac delta distribution [10, 18, 32, 26, 31], which is widely used over past years. As popularized in FCOS [26], predicting an additional localization quality (e.g., IoU score [29] or centerness score [26]) brings consistent improvements of detection accuracy, when

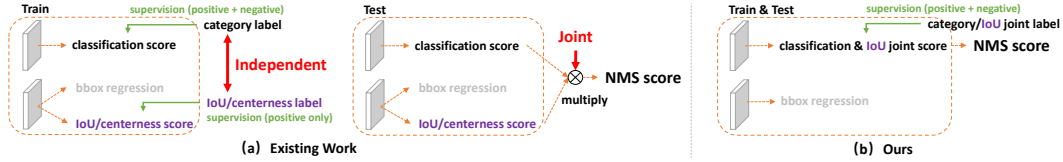

Figure 1: Comparisons between existing separate representation and proposed joint representation of classification and localization quality estimation. (a): Current practices [12, 26, 29, 35, 31] for the separate usage of the quality branch (i.e., IoU or centerness score) during training and test. (b): Our joint representation of classification and localization quality enables high consistency between training and inference.

the quality estimation is combined (usually multiplied) with classification confidence as final scores [12, 11, 26, 29, 35] for the rank process of Non-Maximum Suppression (NMS) during inference. Despite their success, we observe the following problems in existing practices:

**Inconsistent usage of localization quality estimation and classification score between training and inference:** (1) In recent dense detectors, the localization quality estimation and classification score are usually trained independently but compositely utilized (e.g., multiplication) during inference [26, 29] (Fig. 1(a)); (2) The supervision of the localization quality estimation is currently assigned for positive samples only [12, 11, 26, 29, 35], which is unreliable as negatives may get chances to have uncontrollably higher quality predictions (Fig. 2(a)). These two factors result in a gap between training and test, and would potentially degrade the detection performance, e.g., negative instances with randomly high-quality scores could rank in front of positive examples with lower quality prediction during NMS.

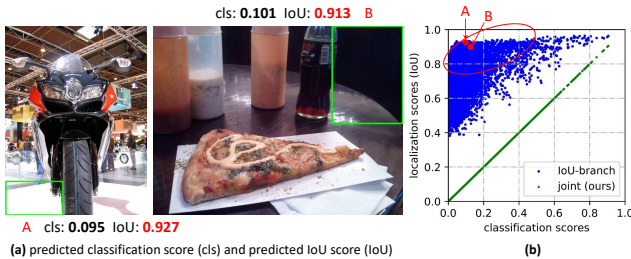

(a) predicted classification score (cls) and predicted IoU score (IoU)

Figure 2: Unreliable IoU predictions of current dense detector with IoU-branch. (a): We demonstrate some background patches (A and B) with extremely high predicted quality scores (e.g., IoU score > 0.9), based on the optimized IoU-branch model in Fig. 1(a). The scatter diagram in (b) denotes the randomly sampled instances with their predicted scores, where the blue points clearly illustrate the weak correlation between predicted classification scores and predicted IoU scores for separate representations. The part in red circle contains many possible negatives with large localization quality predictions, which may potentially rank in front of true positives and impair the performance. Instead, our joint representation (green points) forces them to be equal and thus avoids such risks.

**Inflexible representation of bounding boxes:** The widely used bounding box representation can be viewed as Dirac delta distribution [7, 23, 8, 1, 18, 26, 13, 31] of the target box coordinates. However, it fails to consider the ambiguity and uncertainty in datasets (see the unclear boundaries of the figures in Fig. 3). Although some recent works [10, 4] model boxes as Gaussian distributions, it is too simple to capture the real distribution of the locations of bounding boxes. In fact, the real distribution can be more arbitrary and flexible [10], without the necessity of being symmetric like the Gaussian function.

To address the above problems, we design new representations for the bounding boxes and their localization quality. **For localization quality representation**, we propose to merge it with the classification score into a single and unified representation: a classification vector where its value at the ground-truth category index refers to its corresponding localization quality (typically the IoU score between the predicted box and the corresponding ground-truth box in this paper). In this way, we unify classification score and IoU score into a joint and single variable (denoted as "classification-IoU joint representation"), which can be trained in an end-to-end fashion, whilst directly utilized during inference (Fig. 1(b)). As a result, it eliminates the training-test inconsistency (Fig. 1(b)) and enables the strongest correlation (Fig. 2 (b)) between localization quality and classification. Further, the negatives will be supervised with 0 quality scores, thereby the overall quality predictions become more confidential and reliable. It is especially beneficial for dense object detectors as they rank all candidates regularly sampled across an entire image. **For bounding box representation**, we propose to represent the arbitrary distribution (denoted as "General distribution" in this paper) of box locations by directly learning the discretized probability distribution over its continuous space, without introducing any other stronger priors (e.g., Gaussian [10, 4]). Consequently, we can obtain more reliable and accurate bounding box estimations, whilst being aware of a variety of their underlying distributions (see the predicted distributions in Fig. 3).

The improved representations then pose challenges for optimization. Traditionally for dense detectors, the classification branch is optimized with Focal Loss [18] (FL). FL can successfully handles the

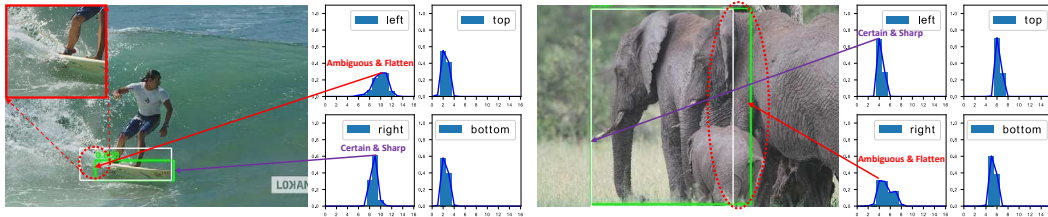

Figure 3: Due to occlusion, shadow, blur, etc., the boundaries of many objects are not clear enough, so that the ground-truth labels (white boxes) are sometimes not credible and Dirac delta distribution is limited to indicate such issues. Instead, the proposed learned representation of General distribution for bounding boxes can reflect the underlying information by its shape, where a flatten distribution denotes the unclear and ambiguous boundaries (see red circles) and a sharp one stands for the clear cases. The predicted boxes by our model are marked green.

class imbalance problem via reshaping the standard cross entropy loss. However, for the case of the proposed classification-IoU joint representation, in addition to the imbalance risk that still exists, we face a new problem with continuous IoU label (0∼1) as supervisions, as the original FL only supports discrete $\{1, 0\}$ category label currently. We successfully solve the problem by extending FL from $\{1, 0\}$ discrete version to its continuous variant, termed Generalized Focal Loss (GFL). Different from FL, GFL considers a much general case in which the globally optimized solution is able to target at any desired continuous value, rather than the discrete ones. More specifically in this paper, GFL can be specialized into Quality Focal Loss (QFL) and Distribution Focal Loss (DFL), for optimizing the improved two representations respectively: QFL focuses on a sparse set of hard examples and simultaneously produces their *continuous* 0∼1 quality estimations on the corresponding category; DFL makes the network to rapidly focus on learning the probabilities of values around the *continuous* locations of target bounding boxes, under an arbitrary and flexible distribution.

We demonstrate three advantages of GFL: (1) It bridges the gap between training and test when one-stage detectors are facilitated with additional quality estimation, leading to a simpler, joint and effective representation of both classification and localization quality; (2) It well models the flexible underlying distribution for bounding boxes, which provides more informative and accurate box locations; (3) The performance of one-stage detectors can be consistently boosted without introducing additional overhead. On COCO `test-dev`, GFL achieves 45.0% AP with ResNet-101 backbone, surpassing state-of-the-art SAPD (43.5%) and ATSS (43.6%). Our best model can achieve a single-model single-scale AP of 48.2% whilst running at 10 FPS on a single 2080Ti GPU.

## 2  Related Work

**Representation of localization quality.** Existing practices like Fitness NMS [27], IoU-Net [12], MS R-CNN [11], FCOS [26] and IoU-aware [29] utilize a separate branch to perform localization quality estimation in a form of IoU or centerness score. As mentioned in Sec. 1, this separate formulation causes the inconsistency between training and test as well as unreliable quality predictions. Instead of introducing an additional branch, PISA [2] and IoU-balance [28] assign different weights in the classification loss based on their localization qualities, aiming at enhancing the correlation between the classification score and localization accuracy. However, the weight strategy is of implicit and limited benefits since it does not change the optimum of the loss objectives for classification.

**Representation of bounding boxes.** Dirac delta distribution [7, 23, 8, 1, 18, 26, 13, 31] governs the representation of bounding boxes over past years. Recently, Gaussian assumption [10, 4] is adopted to learn the uncertainty by introducing a predicted variance. Unfortunately, existing representations are either too rigid or too simplified, which can not reflect the complex underlying distribution in real data. In this paper, we further relax the assumption and directly learn the more arbitrary, flexible General distribution of bounding boxes, whilst being more informative and accurate.

## 3  Method

In this section, we first review the original Focal Loss [18] (FL) for learning dense classification scores of one-stage detectors. Next, we present the details for the improved representations of localization quality estimation and bounding boxes, which are successfully optimized via the proposed Quality Focal Loss (QFL) and Distribution Focal Loss (DFL), respectively. Finally, we summarize the formulations of QFL and DFL into a unified perspective termed Generalized Focal Loss (GFL), as a flexible extension of FL, to facilitate further promotion and general understanding in the future.

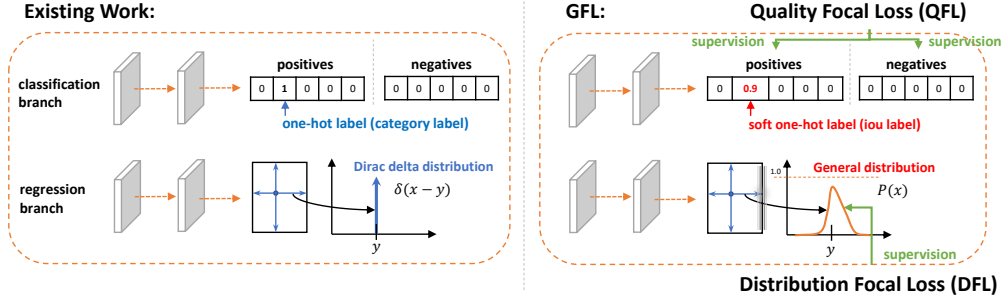

Figure 4: The comparisons between conventional methods and our proposed GFL in the head of dense detectors. GFL includes QFL and DFL. QFL effectively learns a joint representation of classification score and localization quality estimation. DFL models the locations of bounding boxes as General distributions whilst forcing the networks to rapidly focus on learning the probabilities of values close to the target coordinates.

**Focal Loss (FL)**. The original FL [18] is proposed to address the one-stage object detection scenario where an extreme imbalance between foreground and background classes often exists during training. A typical form of FL is as follows (we ignore $\alpha_t$ in original paper [18] for simplicity):

$$\mathbf{FL}(p) = -(1-p_t)^\gamma \log(p_t), p_t = \begin{cases} p, & \text{when } y = 1 \\ 1-p, & \text{when } y = 0 \end{cases} \tag{1}$$

where $y \in \{1, 0\}$ specifies the ground-truth class and $p \in [0, 1]$ denotes the estimated probability for the class with label $y = 1$. $\gamma$ is the tunable focusing parameter. Specifically, FL consists of a standard cross entropy part $-\log(p_t)$ and a dynamically scaling factor part $(1-p_t)^\gamma$, where the scaling factor $(1-p_t)^\gamma$ automatically down-weights the contribution of easy examples during training and rapidly focuses the model on hard examples.

**Quality Focal Loss (QFL)**. To solve the aforementioned inconsistency problem between training and test phases, we present a joint representation of localization quality (i.e., IoU score) and classification score ("classification-IoU" for short), where its supervision softens the standard one-hot category label and leads to a possible float target $y \in [0, 1]$ on the corresponding category (see the classification branch in Fig. 4). Specifically, $y = 0$ denotes the negative samples with 0 quality score, and $0 < y \le 1$ stands for the positive samples with target IoU score $y$. Note that the localization quality label $y$ follows the conventional definition as in [29, 12]: IoU score between the predicted bounding box and its corresponding ground-truth bounding box during training, with a dynamic value being 0~1. Following [18, 26], we adopt the multiple binary classification with sigmoid operators $\sigma(\cdot)$ for multi-class implementation. For simplicity, the output of sigmoid is marked as $\sigma$.

Since the proposed classification-IoU joint representation requires dense supervisions over an entire image and the class imbalance problem still occurs, the idea of FL must be inherited. However, the current form of FL only supports $\{1, 0\}$ discrete labels, but our new labels contain decimals. Therefore, we propose to extend the two parts of FL for enabling the successful training under the case of joint representation: (1) The cross entropy part $-\log(p_t)$ is expanded into its complete version $-\big((1-y)\log(1-\sigma) + y\log(\sigma)\big)$; (2) The scaling factor part $(1-p_t)^\gamma$ is generalized into the absolute distance between the estimation $\sigma$ and its continuous label $y$, i.e., $|y-\sigma|^\beta$ ($\beta \ge 0$), here $|\cdot|$ guarantees the non-negativity. Subsequently, we combine the above two extended parts to formulate the complete loss objective, which is termed as Quality Focal Loss (QFL):

$$\mathbf{QFL}(\sigma) = -|y-\sigma|^\beta \big((1-y)\log(1-\sigma) + y\log(\sigma)\big). \tag{2}$$

Note that $\sigma = y$ is the global minimum solution of QFL. QFL is visualized for several values of $\beta$ in Fig. 5(a) under quality label $y = 0.5$. Similar to FL, the term $|y-\sigma|^\beta$ of QFL behaves as a modulating factor: when the quality estimation of an example is inaccurate and deviated away from label $y$, the modulating factor is relatively large, thus it pays more attention to learning this hard example. As the quality estimation becomes accurate, i.e., $\sigma \to y$, the factor goes to 0 and the loss for well-estimated examples is down-weighted, in which the parameter $\beta$ controls the down-weighting rate smoothly ($\beta = 2$ works best for QFL in our experiments).

**Distribution Focal Loss (DFL).** Following [26, 31], we adopt the relative offsets from the location to the four sides of a bounding box as the regression targets (see the regression branch in Fig. 4). Conventional operations of bounding box regression model the regressed label $y$ as Dirac delta distribution $\delta(x - y)$, where it satisfies $\int_{-\infty}^{+\infty} \delta(x - y)\,\mathrm{d}x = 1$ and is usually implemented through

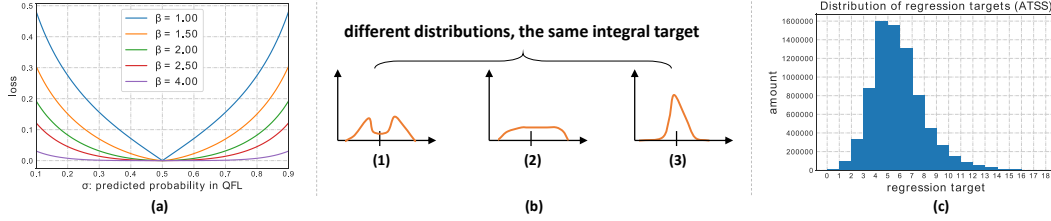

Figure 5: (a): The illustration of QFL under quality label $y = 0.5$. (b): Different flexible distributions can obtain the same integral target according to Eq. (4), thus we need to focus on learning probabilities of values around the target for more reasonable and confident predictions (e.g., (3)). (c): The histogram of bounding box regression targets of ATSS over all training samples on COCO `trainval35k`.

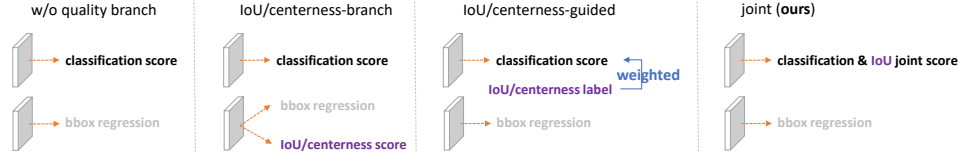

Figure 6: Illustrations of modified versions for separate/implicit and joint representation. The baseline without quality branch is also provided.

fully connected layers. More formally, the integral form to recover $y$ is as follows:

$$y = \int_{-\infty}^{+\infty} \delta(x - y) x \, \mathrm{d}x. \qquad (3)$$

According to the analysis in Sec. 1, instead of the Dirac delta [23, 8, 1, 26, 31] or Gaussian [4, 10] assumptions, we propose to directly learn the underlying General distribution $P(x)$ without introducing any other priors. Given the range of label $y$ with minimum $y_0$ and maximum $y_n$ ($y_0 \le y \le y_n, n \in \mathbb{N}^+$), we can have the estimated value $\hat{y}$ from the model ($\hat{y}$ also meets $y_0 \le \hat{y} \le y_n$):

$$\hat{y} = \int_{-\infty}^{+\infty} P(x) x \, \mathrm{d}x = \int_{y_0}^{y_n} P(x) x \, \mathrm{d}x. \qquad (4)$$

To be consistent with convolutional neural networks, we convert the integral over the continuous domain into a discrete representation, via discretizing the range $[y_0, y_n]$ into a set $\{y_0, y_1, ..., y_i, y_{i+1}, ..., y_{n-1}, y_n\}$ with even intervals $\Delta$, $\Delta = y_{i+1} - y_i, \forall i \in [0, n-1]$ (we use $\Delta = 1$ for simplicity in later experiments). Consequently, given the discrete distribution property $\sum_{i=0}^{n} P(y_i) = 1$, the estimated regression value $\hat{y}$ can be presented as:

$$\hat{y} = \sum_{i=0}^{n} P(y_i) y_i. \qquad (5)$$

As a result, $P(x)$ can be easily implemented through a softmax $\mathcal{S}(\cdot)$ layer consisting of $n+1$ units, with $P(y_i)$ being denoted as $\mathcal{S}_i$ for simplicity. Note that $\hat{y}$ can be trained in an end-to-end fashion with traditional loss objectives like SmoothL1 [7], IoU Loss [27] or GIoU Loss [24]. However, there are infinite combinations of values for $P(x)$ that can make the final integral result being $y$, as shown in Fig. 5(b), which may reduce the learning efficiency. Intuitively compared against (1) and (2), distribution (3) is compact and tends to be more confident and precise on the bounding box estimation, which motivates us to optimize the shape of $P(x)$ via explicitly encouraging the high probabilities of values that are close to the target $y$. Furthermore, it is often the case that the most appropriate underlying location, if exists, would not be far away from the coarse label. Therefore, we introduce the Distribution Focal Loss (DFL) which forces the network to rapidly focus on the values near label $y$, by explicitly enlarging the probabilities of $y_i$ and $y_{i+1}$ (nearest two to $y$, $y_i \le y \le y_{i+1}$). As the learning of bounding boxes are only for positive samples without the risk of class imbalance problem, we simply apply the complete cross entropy part in QFL for the definition of DFL:

$$\mathbf{DFL}(\mathcal{S}_i, \mathcal{S}_{i+1}) = -\big((y_{i+1} - y)\log(\mathcal{S}_i) + (y - y_i)\log(\mathcal{S}_{i+1})\big). \qquad (6)$$

Intuitively, DFL aims to focus on enlarging the probabilities of the values around target $y$ (i.e., $y_i$ and $y_{i+1}$). The global minimum solution of DFL, i.e, $\mathcal{S}_i = \frac{y_{i+1} - y}{y_{i+1} - y_i}, \mathcal{S}_{i+1} = \frac{y - y_i}{y_{i+1} - y_i}$, can guarantee the estimated regression target $\hat{y}$ infinitely close to the corresponding label $y$, i.e., $\hat{y} = \sum_{j=0}^{n} P(y_j) y_j = \mathcal{S}_i y_i + \mathcal{S}_{i+1} y_{i+1} = \frac{y_{i+1} - y}{y_{i+1} - y_i} y_i + \frac{y - y_i}{y_{i+1} - y_i} y_{i+1} = y$, which also ensures its correctness as a loss function.

**Generalized Focal Loss (GFL).** Note that QFL and DFL can be unified into a general form, which is called the Generalized Focal Loss (GFL) in the paper. Assume that a model estimates probabilities

| Type | FCOS [26] | | | | | | ATSS [31] | | | | | |
|---|---|---|---|---|---|---|---|---|---|---|---|---|
| | AP | AP$_{50}$ | AP$_{75}$ | AP$_S$ | AP$_M$ | AP$_L$ | AP | AP$_{50}$ | AP$_{75}$ | AP$_S$ | AP$_M$ | AP$_L$ |
| w/o quality branch | 37.8 | 56.2 | 40.8 | 21.2 | 42.1 | 48.2 | 38.0 | 56.5 | 40.7 | 20.6 | 42.1 | 49.1 |
| centerness-branch [26] | 38.5 | 56.8 | 41.6 | 22.4 | 42.4 | 49.1 | 39.2 | 57.4 | 42.2 | 23.0 | 42.8 | 51.1 |
| IoU-branch [29, 12] | 38.7 | 56.7 | **42.0** | 21.6 | 43.0 | 50.3 | 39.6 | 57.6 | **43.0** | **23.3** | 43.7 | 51.2 |
| centerness-guided [28] | 37.9 | 56.7 | 40.7 | 21.2 | 42.1 | 49.4 | 38.2 | 56.2 | 41.0 | 21.5 | 41.9 | 49.7 |
| IoU-guided [28] | 38.2 | 57.0 | 41.1 | **22.5** | 42.2 | 48.9 | 38.9 | 57.4 | 41.8 | 22.8 | 42.4 | 50.6 |
| joint w/ QFL (ours) | **39.0** | **57.8** | 41.9 | 22.0 | **43.1** | **51.0** | **39.9** | **58.5** | **43.0** | 22.4 | **43.9** | **52.7** |

(a) **Comparisons between separate/implicit and joint representation (ours)**: The joint representation optimized by QFL achieves better performance than other counterparts. We also observe that the quality predictions (especially IoU scores) are necessary for obtaining competitive AP.

| Method | AP | AP$_{50}$ | AP$_{75}$ | AP$_S$ | AP$_M$ | AP$_L$ |
|---|---|---|---|---|---|---|
| FoveaBox [13] | 36.4 | **55.8** | 38.8 | 19.4 | 40.4 | 47.7 |
| FoveaBox [13] + joint w/ QFL | **37.0** | 55.7 | **39.6** | **20.2** | **41.2** | **48.8** |
| RetinaNet [18] | 35.6 | 55.5 | 38.1 | 20.1 | 39.4 | 46.8 |
| RetinaNet [18] + joint w/ QFL | **36.4** | **56.3** | **39.1** | **20.4** | **40.0** | **48.7** |
| SSD512 [20] | 29.4 | 49.1 | 30.6 | 11.4 | 34.1 | 44.9 |
| SSD512 [20] + joint w/ QFL | **30.2** | **50.3** | **31.7** | **13.3** | **34.4** | **45.5** |

| $\beta$ (QFL) | AP | AP$_{50}$ | AP$_{75}$ |
|---|---|---|---|
| 0 | 37.6 | 55.4 | 40.3 |
| 1 | 39.0 | 58.1 | 41.7 |
| 2 | **39.9** | **58.5** | **43.0** |
| 2.5 | 39.7 | 58.1 | 42.7 |
| 4 | 38.2 | 55.4 | 41.6 |

(b) **Applying joint representations with QFL to other one-stage detectors**: About 0.6-0.8 % AP gains are obtained without any additional overhead for inference.

(c) **Varying $\beta$ for QFL based on ATSS**: $\beta = 2$ performs best.

Table 1: Study on QFL (ResNet-50 backbone). All experiments are reproduced in mmdetection [3] and validated on COCO `minival`.

for two variables $y_l, y_r (y_l < y_r)$ as $p_{y_l}, p_{y_r}$ ($p_{y_l} \geq 0, p_{y_r} \geq 0, p_{y_l} + p_{y_r} = 1$), with a final prediction of their linear combination being $\hat{y} = y_l p_{y_l} + y_r p_{y_r} (y_l \leq \hat{y} \leq y_r)$. The corresponding continuous label $y$ for the prediction $\hat{y}$ also satisfies $y_l \leq y \leq y_r$. Taking the absolute distance $|y - \hat{y}|^\beta$ ($\beta \geq 0$) as modulating factor, the specific formulation of GFL can be written as:

$$\mathbf{GFL}(p_{y_l}, p_{y_r}) = -|y - (y_l p_{y_l} + y_r p_{y_r})|^\beta \big((y_r - y) \log(p_{y_l}) + (y - y_l) \log(p_{y_r})\big). \quad (7)$$

**Properties of GFL.** $\mathbf{GFL}(p_{y_l}, p_{y_r})$ reaches its global minimum with $p_{y_l}^* = \frac{y_r - y}{y_r - y_l}, p_{y_r}^* = \frac{y - y_l}{y_r - y_l}$, which also means that the estimation $\hat{y}$ perfectly matches the continuous label $y$, i.e., $\hat{y} = y_l p_{y_l}^* + y_r p_{y_r}^* = y$. Obviously, FL [18] and the proposed QFL and DFL are all *special cases* of GFL:

- **FL**: Letting $\beta = \gamma, y_l = 0, y_r = 1, p_{y_r} = p, p_{y_l} = 1 - p$ and $y \in \{1, 0\}$ in GFL:
$$\mathbf{FL}(p) = \mathbf{GFL}(1 - p, p) = -|y - p|^\gamma \big((1 - y) \log(1 - p) + y \log(p)\big), y \in \{1, 0\}$$
$$= -(1 - p_t)^\gamma \log(p_t), p_t = \begin{cases} p, & \text{when } y = 1 \\ 1 - p, & \text{when } y = 0 \end{cases} \quad (8)$$

- **QFL**: Having $y_l = 0, y_r = 1, p_{y_r} = \sigma$ and $p_{y_l} = 1 - \sigma$ in GFL:
$$\mathbf{QFL}(\sigma) = \mathbf{GFL}(1 - \sigma, \sigma) = -|y - \sigma|^\beta \big((1 - y) \log(1 - \sigma) + y \log(\sigma)\big). \quad (9)$$

- **DFL**: By substituting $\beta = 0, y_l = y_i, y_r = y_{i+1}, p_{y_l} = P(y_l) = P(y_i) = \mathcal{S}_i, p_{y_r} = P(y_r) = P(y_{i+1}) = \mathcal{S}_{i+1}$ in GFL:
$$\mathbf{DFL}(\mathcal{S}_i, \mathcal{S}_{i+1}) = \mathbf{GFL}(\mathcal{S}_i, \mathcal{S}_{i+1}) = -\big((y_{i+1} - y) \log(\mathcal{S}_i) + (y - y_i) \log(\mathcal{S}_{i+1})\big). \quad (10)$$

Note that GFL can be applied to any one-stage detectors. The modified detectors differ from the original detectors in two aspects. First, during inference, we directly feed the classification score (joint representation with quality estimation) as NMS scores without the need of multiplying any *individual* quality prediction if there exists (e.g., centerness as in FCOS [26] and ATSS [31]). Second, the last layer of the regression branch for predicting each location of bounding boxes now has $n + 1$ outputs instead of 1 output, which brings *negligible* extra computing cost as later shown in Table 3.

**Training Dense Detectors with GFL.** We define training loss $\mathcal{L}$ with GFL:

$$\mathcal{L} = \frac{1}{N_{pos}} \sum_z \mathcal{L}_Q + \frac{1}{N_{pos}} \sum_z \mathbf{1}_{\{c_z^* > 0\}} \big(\lambda_0 \mathcal{L}_\mathcal{B} + \lambda_1 \mathcal{L}_\mathcal{D}\big), \quad (11)$$

where $\mathcal{L}_Q$ is QFL and $\mathcal{L}_\mathcal{D}$ is DFL. Typically, $\mathcal{L}_\mathcal{B}$ denotes the GIoU Loss as in [26, 31]. $N_{pos}$ stands for the number of positive samples. $\lambda_0$ (typically 2 as default, similarly in [3]) and $\lambda_1$ (practically $\frac{1}{4}$, averaged over four directions) are the balance weights for $\mathcal{L}_Q$ and $\mathcal{L}_\mathcal{D}$, respectively. The summation is calculated over all locations $z$ on the pyramid feature maps [17]. $\mathbf{1}_{\{c_z^* > 0\}}$ is the indicator function, being 1 if $c_z^* > 0$ and 0 otherwise. Following the common practices in the official codes [3, 26, 31, 15], we also utilize the quality scores to weight $\mathcal{L}_\mathcal{B}$ and $\mathcal{L}_\mathcal{D}$ during training.

| Prior Distribution | FCOS [26] | | | | | | ATSS [31] | | | | | |
|---|---|---|---|---|---|---|---|---|---|---|---|---|
| | AP | AP$_{50}$ | AP$_{75}$ | AP$_S$ | AP$_M$ | AP$_L$ | AP | AP$_{50}$ | AP$_{75}$ | AP$_S$ | AP$_M$ | AP$_L$ |
| Dirac delta [26, 31] | 38.5 | 56.8 | 41.6 | 22.4 | 42.4 | 49.1 | 39.2 | **57.4** | 42.2 | 23.0 | 42.8 | 51.1 |
| Gaussian [10, 4] | 38.6 | 56.5 | 41.6 | 21.7 | 42.5 | 50.0 | 39.3 | 57.0 | 42.4 | **23.6** | 42.9 | 51.0 |
| General (**ours**) | 38.8 | 56.6 | 42.0 | 22.5 | 42.9 | 49.8 | 39.3 | 57.1 | 42.5 | 23.5 | 43.0 | **51.2** |
| General w/ DFL (**ours**) | **39.0** | **57.0** | **42.3** | **22.6** | **43.0** | 50.6 | **39.5** | 57.3 | **42.8** | 23.6 | **43.2** | 51.2 |

(a) **Performances under different data representation of bounding box regression targets**: the proposed General distribution supervised by DFL improves favorably over the competitive baselines.

| n | $\Delta$ | AP | AP$_{50}$ | AP$_{75}$ | AP$_S$ | AP$_M$ | AP$_L$ |
|---|---|---|---|---|---|---|---|
| 12 | | 40.1 | 58.4 | 43.1 | 23.1 | 43.8 | 52.5 |
| 14 | | **40.2** | 58.3 | **43.6** | **23.3** | 44.2 | 52.2 |
| 16 | 1 | **40.2** | **58.6** | 43.4 | 23.0 | **44.3** | **53.0** |
| 18 | | 40.1 | 58.1 | 43.1 | 22.6 | 43.9 | 52.6 |

| $y_n$ | $\Delta$ | AP | AP$_{50}$ | AP$_{75}$ | AP$_S$ | AP$_M$ | AP$_L$ |
|---|---|---|---|---|---|---|---|
| | 0.5 | **40.2** | 58.4 | 43.0 | 22.3 | 43.8 | **53.1** |
| 16 | 1 | **40.2** | **58.6** | **43.4** | **23.0** | **44.3** | 53.0 |
| | 2 | 39.9 | 58.3 | 42.9 | 22.5 | 43.8 | 51.8 |
| | 4 | 39.8 | 58.5 | 42.8 | 22.8 | 43.4 | 52.3 |

(b) **Varying $n$ by fixing $\Delta = 1$ on ATSS (w/ GFL)**: The performance is robust to a range of $n$ according to its target distribution in Fig. 5(c).

(c) **Varying $\Delta$ by fixing $y_n = 16$ on ATSS (w/ GFL)**: Small $\Delta$ usually leads to better performance whilst $\Delta = 1$ is good enough for practice.

Table 2: Study on DFL (ResNet-50 backbone). All experiments are reproduced in mmdetection [3] and validated on COCO `minival`.

# 4 Experiment

Our experiments are conducted on COCO benchmark [19], where `trainval35k` (115K images) is utilized for training and we use `minival` (5K images) as validation for our ablation study. The main results are reported on `test-dev` (20K images) which can be obtained from the evaluation server. For fair comparisons, all results are produced under mmdetection [3], where the default hyper-parameters are adopted. Unless otherwise stated, we adopt 1x learning schedule (12 epochs) without multi-scale training for the following studies, based on ResNet-50 [9] backbone.

We first investigate the effectiveness of the QFL (Table 1). In Table 1(a), we compare the proposed joint representation with its separate or implicit counterparts. Two alternatives for representing localization quality: IoU [29, 12] and centerness [26, 31] are also adopted in the experiments. In general, we construct 4 variants that use separate or implicit representation, as illustrated in Fig. 6. According to the results, we observe that the joint representations optimized by QFL consistently achieve better performance than all the counterparts, whilst IoU always performs better than centerness as a measurement of localization quality (see more analyses in Sec. 5). Table 1(b) shows that QFL can also boost the performance of other popular one-stage detectors, and Table 1(c) shows that $\beta = 2$ is the best setting for QFL. We illustrate the effectiveness of joint representation by sampling instances with its predicted classification and IoU scores of both IoU-branch model and ours, as shown in Fig. 2(b). It demonstrates that the proposed joint representation trained with QFL can benefit the detection due to its more reliable quality estimation, and yields the strongest correlation between classification and quality scores according to its definition. In fact, in our joint representation, the predicted classification score is equal to the estimated quality score exactly.

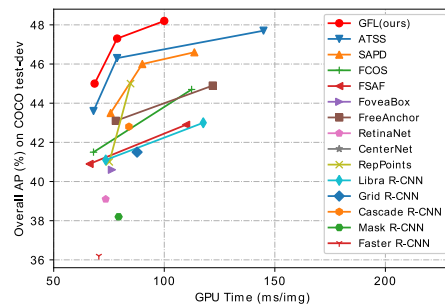

Figure 7: Single-model single-scale speed (ms) vs. accuracy (AP) on COCO test-dev among state-of-the-art approaches. GFL achieves better speed-accuracy trade-off than many competitive counterparts.

| QFL | DFL | FPS | AP | AP$_{50}$ | AP$_{75}$ |
|---|---|---|---|---|---|
| | | 19.4 | 39.2 | 57.4 | 42.2 |
| ✓ | | 19.4 | 39.9 | 58.5 | 43.0 |
| | ✓ | 19.4 | 39.5 | 57.3 | 42.8 |
| ✓ | ✓ | 19.4 | **40.2** | **58.6** | **43.4** |

Table 3: **The effect of QFL and DFL on ATSS**: The effects of QFL and DFL are orthogonal, whilst utilizing both can boost 1% AP over the strong ATSS baseline, without introducing additional overhead practically.

Second, we investigate the effectiveness of the DFL (Table 2). To quickly select a reasonable value of $n$, we first illustrate the distribution of the regression targets in Fig. 5(c). We will show in later experiments, the recommended choice of $n$ for ATSS is 14 or 16. In Table 2(a), we compare the effectiveness of different data representations for bounding box regression. We find that the General distribution achieves superior or at least comparable results, whilst DFL can further boost its performance. Based on the improved ATSS trained by GFL, we report the effect of $n$ and $\Delta$ in DFL in Table 2(b) and (c). The results demonstrate that the selection of $n$ is not sensitive and $\Delta$ is suggested to be small (e.g., 1) in practice. To illustrate the effect of General distribution, we

| Method | Backbone | Epoch | MS$_{train}$ | FPS | AP | AP$_{50}$ | AP$_{75}$ | AP$_S$ | AP$_M$ | AP$_L$ | Reference |
|---|---|---|---|---|---|---|---|---|---|---|---|
| *multi-stage:* | | | | | | | | | | | |
| Faster R-CNN w/ FPN [17] | R-101 | 24 | | 14.2 | 36.2 | 59.1 | 39.0 | 18.2 | 39.0 | 48.2 | CVPR17 |
| Cascade R-CNN [1] | R-101 | 18 | | 11.9 | 42.8 | 62.1 | 46.3 | 23.7 | 45.5 | 55.2 | CVPR18 |
| Grid R-CNN [21] | R-101 | 20 | | 11.4 | 41.5 | 60.9 | 44.5 | 23.3 | 44.9 | 53.1 | CVPR19 |
| Libra R-CNN [22] | R-101 | 24 | | 13.6 | 41.1 | 62.1 | 44.7 | 23.4 | 43.7 | 52.5 | CVPR19 |
| Libra R-CNN [22] | X-101-64x4d | 12 | | 8.5 | 43.0 | 64.0 | 47.0 | 25.3 | 45.6 | 54.6 | CVPR19 |
| RepPoints [30] | R-101 | 24 | | 13.3 | 41.0 | 62.9 | 44.3 | 23.6 | 44.1 | 51.7 | ICCV19 |
| RepPoints [30] | R-101-DCN | 24 | ✓ | 11.8 | 45.0 | 66.1 | 49.0 | 26.6 | 48.6 | 57.5 | ICCV19 |
| TridentNet [16] | R-101 | 24 | ✓ | 2.7* | 42.7 | 63.6 | 46.5 | 23.9 | 46.6 | 56.6 | ICCV19 |
| TridentNet [16] | R-101-DCN | 36 | ✓ | 1.3* | 46.8 | 67.6 | 51.5 | 28.0 | 51.2 | 60.5 | ICCV19 |
| TSD [25] | R-101 | 20 | | 1.1 | 43.2 | 64.0 | 46.9 | 24.0 | 46.3 | 55.8 | CVPR20 |
| *one-stage:* | | | | | | | | | | | |
| CornerNet [14] | HG-104 | 200 | ✓ | 3.1* | 40.6 | 56.4 | 43.2 | 19.1 | 42.8 | 54.3 | ECCV18 |
| CenterNet [6] | HG-52 | 190 | ✓ | 4.4* | 41.6 | 59.4 | 44.2 | 22.5 | 43.1 | 54.1 | ICCV19 |
| CenterNet [6] | HG-104 | 190 | ✓ | 3.3* | 44.9 | 62.4 | 48.1 | 25.6 | 47.4 | 57.4 | ICCV19 |
| CentripetalNet [5] | HG-104 | 210 | ✓ | n/a | 45.8 | 63.0 | 49.3 | 25.0 | 48.2 | 58.7 | CVPR20 |
| RetinaNet [18] | R-101 | 18 | | 13.6 | 39.1 | 59.1 | 42.3 | 21.8 | 42.7 | 50.2 | ICCV17 |
| FreeAnchor [32] | R-101 | 24 | ✓ | 12.8 | 43.1 | 62.2 | 46.4 | 24.5 | 46.1 | 54.8 | NeurIPS19 |
| FreeAnchor [32] | X-101-32x8d | 24 | ✓ | 8.2 | 44.9 | 64.3 | 48.5 | 26.8 | 48.3 | 55.9 | NeurIPS19 |
| FoveaBox [13] | R-101 | 18 | ✓ | 13.1 | 40.6 | 60.1 | 43.5 | 23.3 | 45.2 | 54.5 | – |
| FoveaBox [13] | X-101 | 18 | ✓ | n/a | 42.1 | 61.9 | 45.2 | 24.9 | 46.8 | 55.6 | – |
| FSAF [34] | R-101 | 18 | ✓ | 15.1 | 40.9 | 61.5 | 44.0 | 24.0 | 44.2 | 51.3 | CVPR19 |
| FSAF [34] | X-101-64x4d | 18 | ✓ | 9.1 | 42.9 | 63.8 | 46.3 | 26.6 | 46.2 | 52.7 | CVPR19 |
| FCOS [26] | R-101 | 24 | ✓ | 14.7 | 41.5 | 60.7 | 45.0 | 24.4 | 44.8 | 51.6 | ICCV19 |
| FCOS [26] | X-101-64x4d | 24 | ✓ | 8.9 | 44.7 | 64.1 | 48.4 | 27.6 | 47.5 | 55.6 | ICCV19 |
| SAPD [33] | R-101 | 24 | ✓ | 13.2 | 43.5 | 63.6 | 46.5 | 24.9 | 46.8 | 54.6 | CVPR20 |
| SAPD [33] | X-101-32x4d | 24 | ✓ | 10.7 | 44.5 | 64.7 | 47.8 | 26.5 | 47.8 | 55.8 | CVPR20 |
| SAPD [33] | R-101-DCN | 24 | ✓ | 11.1 | 46.0 | 65.9 | 49.6 | 26.3 | 49.2 | 59.6 | CVPR20 |
| SAPD [33] | X-101-32x4d-DCN | 24 | ✓ | 8.8 | 46.6 | 66.6 | 50.0 | 27.3 | 49.7 | 60.7 | CVPR20 |
| ATSS [31] | R-101 | 24 | ✓ | 14.6 | 43.6 | 62.1 | 47.4 | 26.1 | 47.0 | 53.6 | CVPR20 |
| ATSS [31] | X-101-32x8d | 24 | ✓ | 8.9 | 45.1 | 63.9 | 49.1 | 27.9 | 48.2 | 54.6 | CVPR20 |
| ATSS [31] | R-101-DCN | 24 | ✓ | 12.7 | 46.3 | 64.7 | 50.4 | 27.7 | 49.8 | 58.4 | CVPR20 |
| ATSS [31] | X-101-32x8d-DCN | 24 | ✓ | 6.9 | 47.7 | 66.6 | 52.1 | 29.3 | 50.8 | 59.7 | CVPR20 |
| GFL (**ours**) | R-50 | 24 | ✓ | 19.4 | 43.1 | 62.0 | 46.8 | 26.0 | 46.7 | 52.3 | – |
| GFL (**ours**) | R-101 | 24 | ✓ | 14.6 | 45.0 | 63.7 | 48.9 | 27.2 | 48.8 | 54.5 | – |
| GFL (**ours**) | X-101-32x4d | 24 | ✓ | 12.2 | 46.0 | 65.1 | 50.1 | 28.2 | 49.6 | 56.0 | – |
| GFL (**ours**) | R-101-DCN | 24 | ✓ | 12.7 | 47.3 | 66.3 | 51.4 | 28.0 | 51.1 | 59.2 | – |
| GFL (**ours**) | X-101-32x4d-DCN | 24 | ✓ | 10.0 | 48.2 | 67.4 | 52.6 | 29.2 | 51.7 | 60.2 | – |

Table 4: Comparisons between state-of-the-art detectors *(single-model and single-scale results)* on COCO `test-dev`. "MS$_{train}$" denotes multi-scale training. FPS values with * are from [33], while others are measured on the same machine with a single GeForce RTX 2080Ti GPU under the same mmdetection [3] framework, using a batch size of 1 whenever possible. "n/a" means that both trained models and timing results from original papers are not available. **R**: ResNet. **X**: ResNeXt. **HG**: Hourglass. **DCN**: Deformable Convolutional Network.

plot several representative instances with its distributed bounding box over four directions in Fig. 3, where the proposed distributed representation can effectively reflect the uncertainty of bounding boxes by its shape.

Third, we perform the ablation study on ATSS with ResNet-50 backbone to show the relative contributions of QFL and DFL (Table 3). FPS (Frames-per-Second) is measured on the same machine with a single GeForce RTX 2080Ti GPU using a batch size of 1 under the same mmdetection [3] framework. We observe that the improvement of DFL is orthogonal to QFL, and joint usage of both (i.e., GFL) improves the strong ATSS baseline by absolute 1% AP score. Furthermore, according to the inference speeds, GFL brings negligible additional overhead and is considered very practical.

Finally, we compare GFL (based on ATSS) with state-of-the-art approaches on COCO `test-dev` in Table 4. Following previous works [18, 26], the multi-scale training strategy and 2x learning schedule (24 epochs) are adopted during training. For a fair comparison, we report the results of single-model single-scale testing for all methods, as well as their corresponding inference speeds (FPS). GFL with ResNet-101 [9] achieves 45.0% AP at 14.6 FPS, which is superior than all the existing detectors with the same backbone, including SAPD [33] (43.5%) and ATSS [31] (43.6%). Further, Deformable Convolutional Networks (DCN) [36] consistently boost the performances over ResNe(X)t backbones, where GFL with ResNeXt-101-32x4d-DCN obtains state-of-the-art 48.2% AP at 10 FPS. Fig. 7 demonstrates the visualization of the accuracy-speed trade-off, where it can be observed that our proposed GFL pushes the envelope of accuracy-speed boundary to a high level.

# 5 Analysis

The ablation study Table 1 also demonstrates that for FCOS/ATSS, IoU performs consistently better than centerness, as a measurement of localization quality. Here we give a convincing reason why this is the case. We discover the major problem of centerness is that its definition leads to unexpected small ground-truth label, which makes a possible set of ground-truth bounding boxes extremely hard to be recalled (as shown in Fig. 8). From the label distributions demonstrated in Fig. 9, we observe that most of IoU labels is larger than 0.4 yet centerness labels tend to be much smaller (even approaching 0). The small values of centerness labels prevent a set of ground-truth bounding boxes from being recalled, as their final scores for NMS would be potentially small since their predicted centerness scores are already supervised by these extremely small signals.

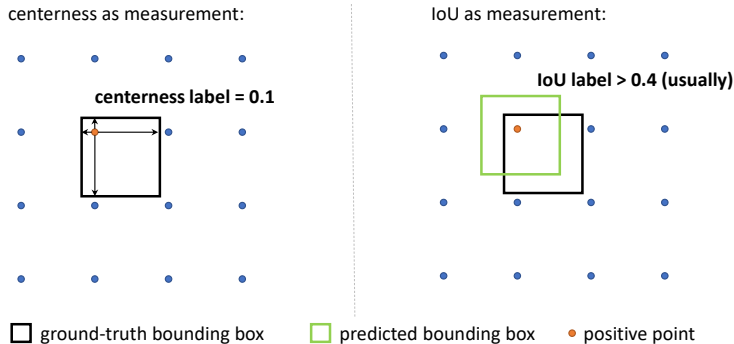

Figure 8: We demonstrate possible cases of ground-truth/predicted bounding box along with the positive points. The matrix points denote the feature pyramid layer with stride = 8. Centerness label is easier to get very small values by its definition, whilst IoU label is more reliable as the supervisions from bounding boxes will always push it close to 1.0.

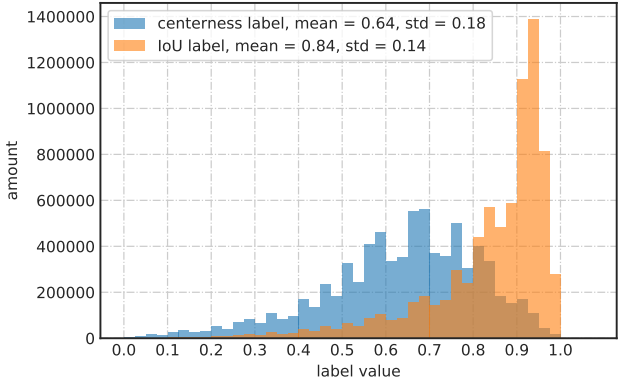

Figure 9: Label distributions over all positive training samples on COCO, based on pretrained GFL detector (ResNet-50 backbone).

# 6 Conclusion

To effectively learn qualified and distributed bounding boxes for dense object detectors, we propose Generalized Focal Loss (GFL) that generalizes the original Focal Loss from $\{1, 0\}$ discrete formulation to the continuous version. GFL can be specialized into Quality Focal loss (QFL) and Distribution Focal Loss (DFL), where QFL encourages to learn a better joint representation of classification and localization quality, and DFL provides more informative and precise bounding box estimations by modeling their locations as General distributions. We also provide a convincing reason and suggest that the community should use IoU instead of centerness as quality measurement although centerness is very successful in FCOS and ATSS. Extensive experiments validate the effectiveness of GFL. We hope GFL can serve as a simple yet effective baseline for the community.

## Broader Impact

Superior performances for object detection tasks indeed have some societal consequences. Specifically for our work, we push the boundary of accuracy-speed for dense detectors to a new level by generating a fast and also accurate object detector. The improved detector can have benefits for a range of fields that involves object recognition tasks, e.g., vision-based self-driving or visual navigation, to avoid accidents and ensure human safety. Further, the ideas of (1) improving representations via a joint formulation for classification and localization quality and (2) directly learning the arbitrary distribution of box locations potentially demonstrate many insights and can inspire more thinking on the representation learning of the computer vision community. Finally, the technique in this paper has no obvious negative ethical and harmful social impact.

## Acknowledgments and Disclosure of Funding

The authors would like to thank the editor and the anonymous reviewers for their critical and constructive comments and suggestions. This work was supported by Postdoctoral Innovative Talent Support Program of China under Grant BX20200168, 2020M681608, NSFC 62072242, 61836014, U19B2034 and U1713208, Program for Changjiang Scholars.

## Footnotes

*Corresponding author. Xiang Li, Jun Li and Jian Yang are from PCA Lab, Key Lab of Intelligent Perception and Systems for High-Dimensional Information of Ministry of Education, and Jiangsu Key Lab of Image and Video Understanding for Social Security, School of Computer Science and Engineering, Nanjing University of Science and Technology. Xiang Li is also a visiting scholar at Momenta.

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
