[Supplementary Material · GFL_SM.pdf]

## A  More Discussions about the Distributions

Fig. 9 depicts the ideas of Dirac delta, Gaussian, and the proposed General distributions, where the assumption goes from rigid (Dirac delta) to flexible (General). We also list several key comparisons about these distributions in Table 5. It can be observed that the loss objective of the Gaussian assumption is actually a dynamically weighted L2 Loss, where its training weight is related to the predicted variance $\sigma$. It is somehow similar to that of Dirac delta (standard L2 Loss) when optimized at the edge level. Moreover, it is not clear how to integrate the Gaussian assumption into the IoU-based Loss formulations, since it heavily couples the expression of the target representation with its optimization objective. Therefore, it can not enjoy the benefits of the IoU-based optimization [24], as it is proved to be very effective in practice. In contrast, our proposed General distribution decouples the representation and loss objective, making it feasible for any type of optimizations, including both edge level and box level.

Figure 9: Illustrations of three distributions, from rigid (Dirac delta) to flexible (General). The proposed General distribution is more flexible as its shape can be arbitrary. In contrast, Dirac delta distribution roots at a fixed point and Gaussian distribution follows a relatively rigid, symmetric expression, e.g., $\frac{1}{\sigma\sqrt{2\pi}}e^{-\frac{(x-\mu)^2}{2\sigma^2}}$, which both have more limitations in modeling real data distribution.

| Type | Dirac delta [26, 31] | | Gaussian [4, 10] | General (ours) | |
|---|---|---|---|---|---|
| Probability Density | $\delta(x-y)$ | | $N(x,\sigma^2)$ | $P(x)$ | |
| Inference Target | $x$ | | $x$ | $\int P(x)x\,\mathrm{d}x$ | |
| Loss Objective (for box part) | $\frac{(x-y)^2}{2}$ | IoU-based Loss | $\frac{(x-y)^2}{2\sigma^2}+\frac{1}{2}\log(\sigma^2)$ | $\frac{(\int P(x)x\,\mathrm{d}x-y)^2}{2}$ | IoU-based Loss |
| Optimization Level | edge | box | edge | edge | box |

Table 5: Comparisons between three distributions. "edge" level denotes optimization over four respective directions, whilst "box" level means IoU-based Losses [24] that consider the bounding box as a whole.

Figure 10: We demonstrate an example in 2D space by fixing the input feature vector and introduce a small disturbance (norm of 0.1) over it. The regression targets are 1.5, 2.5, 3.5 respectively. It is observed that Dirac delta distribution leads to more regression errors after the same disturbance, and the error increases with the growth of regression target. In contrast, our proposed General distribution remains stable and insensitive to the disturbance.

We also find that the bounding box regression of Dirac delta distribution (including Gaussian distribution based on the analysis from Table 5) behaves more sensitive to feature perturbations, making it less robust and susceptible to noise, as shown in the simulation experiment (Fig. 10). It proves that General distribution enjoys more benefits than the other counterparts.

## B  Global Minimum of $\mathbf{GFL}(p_{y_l}, p_{y_r})$

Let's review the definition of **GFL**:

$$\mathbf{GFL}(p_{y_l}, p_{y_r}) = -\left|y - (y_l p_{y_l} + y_r p_{y_r})\right|^\beta \big((y_r - y)\log(p_{y_l}) + (y - y_l)\log(p_{y_r})\big), \quad \text{given } p_{y_l} + p_{y_r} = 1.$$

378    For simplicity, $\mathbf{GFL}(p_{y_l}, p_{y_r})$ can then be expanded as:

$$\mathbf{GFL}(p_{y_l}, p_{y_r}) = -\left|y - (y_l p_{y_l} + y_r p_{y_r})\right|^{\beta}\left((y_r - y)\log(p_{y_l}) + (y - y_l)\log(p_{y_r})\right)$$

$$= \underbrace{\left\{\left|y - (y_l p_{y_l} + y_r p_{y_r})\right|^{\beta}\right\}}_{\mathbf{L}(\cdot,\cdot)} \underbrace{\left\{-\left((y_r - y)\log(p_{y_l}) + (y - y_l)\log(p_{y_r})\right)\right\}}_{\mathbf{R}(\cdot,\cdot)}$$

$$= \mathbf{L}(p_{y_l}, p_{y_r})\mathbf{R}(p_{y_l}, p_{y_r}),$$

$$\mathbf{R}(p_{y_l}, p_{y_r}) = -\left((y_r - y)\log(p_{y_l}) + (y - y_l)\log(p_{y_r})\right)$$

$$= -\left((y_r - y)\log(p_{y_l}) + (y - y_l)\log(1 - p_{y_l})\right)$$

$$\geq -\left((y_r - y)\log(\frac{y_r - y}{y_r - y_l}) + (y - y_l)\log(\frac{y - y_l}{y_r - y_l})\right)$$

$$= \mathbf{R}(p_{y_l}^*, p_{y_r}^*) > 0, \quad \text{where} \quad p_{y_l}^* = \frac{y_r - y}{y_r - y_l}, p_{y_r}^* = \frac{y - y_l}{y_r - y_l}.$$

$$\mathbf{L}(p_{y_l}, p_{y_r}) = \left|y - (y_l p_{y_l} + y_r p_{y_r})\right|^{\beta}$$

$$\geq \mathbf{L}(p_{y_l}^*, p_{y_r}^*) = 0, \quad \text{where} \quad p_{y_l}^* = \frac{y_r - y}{y_r - y_l}, p_{y_r}^* = \frac{y - y_l}{y_r - y_l}.$$

379    Furthermore, given $\epsilon \neq 0$, for arbitrary variable $(p_{y_l}, p_{y_r}) = (p_{y_l}^* + \epsilon, p_{y_r}^* - \epsilon)$ in the domain of definition, we
380    can have:

$$\mathbf{R}(p_{y_l}, p_{y_r}) = \mathbf{R}(p_{y_l}^* + \epsilon, p_{y_r}^* - \epsilon) > \mathbf{R}(p_{y_l}^*, p_{y_r}^*) > 0,$$

$$\mathbf{L}(p_{y_l}, p_{y_r}) = \mathbf{L}(p_{y_l}^* + \epsilon, p_{y_r}^* - \epsilon) = \left|\epsilon(y_r - y_l)\right|^{\beta} > 0 = \mathbf{L}(p_{y_l}^*, p_{y_r}^*).$$

381    Therefore, it is easy to deduce:

$$\mathbf{GFL}(p_{y_l}, p_{y_r}) = \mathbf{L}(p_{y_l}, p_{y_r})\mathbf{R}(p_{y_l}, p_{y_r}) \geq \mathbf{L}(p_{y_l}^*, p_{y_r}^*)\mathbf{R}(p_{y_l}^*, p_{y_r}^*) = 0,$$

382    where "=" holds only when $p_{y_l} = p_{y_l}^*, p_{y_r} = p_{y_r}^*$.

383    The global minimum property of GFL somehow explains why the IoU or centerness guided variants in Fig. 6
384    would not have obvious advantages. In fact, the weighted guidance does not essentially change the global
385    minimum of the original classification loss (e.g., Focal Loss), whilst their optimal classification targets are still
386    one-hot labels. In contrast, the proposed GFL indeed modifies the global minimum and force the predictions to
387    approach the accurate IoU between the estimated boxes and ground-truth boxes, which is obviously beneficial
388    for the rank process of NMS.

## 389   C   FL, QFL and DFL are special cases of GFL

390    In this section, we show how GFL can be specialized into the form of FL, QFL and DFL, respectively.

391    **FL**: Letting $\beta = \gamma, y_l = 0, y_r = 1, p_{y_r} = p, p_{y_l} = 1 - p$ and $y \in \{1, 0\}$ in GFL, we can obtain FL:

$$\mathbf{FL}(p) = \mathbf{GFL}(1 - p, p) = -\left|y - p\right|^{\gamma}\left((1 - y)\log(1 - p) + y\log(p)\right), y \in \{1, 0\}$$

$$= -(1 - p_t)^{\gamma}\log(p_t), p_t = \begin{cases} p, & \text{when } y = 1 \\ 1 - p, & \text{when } y = 0 \end{cases} \tag{9}$$

392    **QFL**: Having $y_l = 0, y_r = 1, p_{y_r} = \sigma$ and $p_{y_l} = 1 - \sigma$ in GFL, the form of QFL can be written as:

$$\mathbf{QFL}(\sigma) = \mathbf{GFL}(1 - \sigma, \sigma) = -\left|y - \sigma\right|^{\beta}\left((1 - y)\log(1 - \sigma) + y\log(\sigma)\right). \tag{10}$$

393    **DFL**: By substituting $\beta = 0, y_l = y_i, y_r = y_{i+1}, p_{y_l} = P(y_l) = P(y_i) = \mathcal{S}_i, p_{y_r} = P(y_r) = P(y_{i+1}) =$
394    $\mathcal{S}_{i+1}$ in GFL, we can have DFL:

$$\mathbf{DFL}(\mathcal{S}_i, \mathcal{S}_{i+1}) = \mathbf{GFL}(\mathcal{S}_i, \mathcal{S}_{i+1}) = -\left((y_{i+1} - y)\log(\mathcal{S}_i) + (y - y_i)\log(\mathcal{S}_{i+1})\right). \tag{11}$$

## 395   D   Details of Experimental Settings

396    **Training Details:** The ImageNet pretrained models [9] with FPN [17] are utilized as the backbones. During
397    training, the input images are resized to keep their shorter side being 800 and their longer side less or equal to
398    1333. In ablation study, the networks are trained using the Stochastic Gradient Descent (SGD) algorithm for
399    90K iterations (denoted as 1x schedule) with 0.9 momentum, 0.0001 weight decay and 16 batch size. The initial
400    learning rate is set as 0.01 and decayed by 0.1 at iteration 60K and 80K, respectively.

401    **Inference Details:** During inference, the input image is resized in the same way as in the training phase, and
402    then passed through the whole network to output the predicted bounding boxes with a predicted class. Then we

use the threshold 0.05 to filter out a variety of backgrounds, and output top 1000 candidate detections per feature
pyramid. Finally, NMS is applied under the IoU threshold 0.6 per class to produce the final top 100 detections
per image as results.

# E    Why is IoU-branch always superior than centerness-branch?

The ablation study in original paper also demonstrates that for FCOS/ATSS, IoU performs consistently better
than centerness, as a measurement of localization quality. Here we give a convincing reason why this is the case.
We discover the major problem of centerness is that its definition leads to unexpected small ground-truth label,
which makes a possible set of ground-truth bounding boxes extremely hard to be recalled (as shown in Fig. 11).
From the label distributions demonstrated in Fig. 12, we observe that most of IoU labels is larger than 0.4 yet
centerness labels tend to be much smaller (even approaching 0). The small values of centerness labels prevent a
set of ground-truth bounding boxes from being recalled, as their final scores for NMS would be potentially small
since their predicted centerness scores are already supervised by these extremely small signals.

Figure 11: We demonstrate possible cases of ground-truth/predicted bounding box along with the positive points. The matrix points denote the
feature pyramid layer with stride = 8. Centerness label is easier to get very small values by its definition, whilst IoU label is more reliable as
the supervisions from bounding boxes will always push it close to 1.0.

Figure 12: Label distributions over all positive training samples on COCO, based on pretrained GFL detector (ResNet-50 backbone).

# F    More Examples of Distributed Bounding Boxes

We demonstrate more examples with General distributed bounding boxes predicted by GFL (ResNet-50 back-
bone). As demonstrated in Fig. 13, we show several cases with boundary ambiguities: does the slim and almost
invisible backpack strap belong to the box of the bag (left top)? does the partially occluded umbrella handle
belong to the entire umbrella (left middle)? In these cases, our models even produce more reasonable coordinates
of bounding boxes than the ground-truth ones. In Fig. 14, more examples with clear boundaries and sharp
General distributions are shown, where GFL is very confident to generate accurate bounding boxes, e.g., the
bottom parts of the orange and the skiing woman.

Figure 13: Examples with huge boundary ambiguities and uncertainties, where the learned General distributions tend to be flatten. In some cases, we even observe a distribution with two peaks. Interestingly, they do correspond to two different most likely boundaries in the image, e.g., the boundaries of the umbrella whether its heavily occluded handle is considered. Predictions are marked green in images, whilst ground-truth boxes are white.

Figure 14: Examples with extremely clear boundaries. The learned General distributions are relatively sharp whilst producing very accurate box estimations. Predictions are marked green in images, whilst ground-truth boxes are white.