[Reviews · NeurIPS 2020]

Review 1

Summary and Contributions: This paper identifies two problems in existing detectors. First, the current loss function for classification does not take the quality of a prediction (whether it sufficiently overlaps with ground-truth) into account. Second, the current loss function for regression does not allow ambiguity and uncertainty in a complex scene because of the inflexible Dirac delta distribution assumption which assumes only one correct value. The authors propose an IoU loss function which trains the network to estimate how well a prediction overlaps with the ground-truth. They also propose a regression function which does away the Dirac delta distribution assumption, and allow the network to learn the probabilities of values around the ground-truth value. The authors design both the loss functions by extending the focal loss function.

Strengths: I like the idea that trains the network to directly predict the overlaps between the bounding boxes and ground-truths. This better fit the post-processing step such as non-maximal suppression commonly found in modern state-of-the-art object detectors. The authors clearly show that how they derive the generalized version of focal loss to be used in training their network. The authors show that the proposed loss function can improve the performance of ATSS from 47.7% to 48.2% on the COCO test-dev dataset.

Weaknesses: The idea of predicting IoUs between predictions and ground-truths has already been explored by other works. It seems to me the main differences between this and prior works are: 1) Prior works such as IoU-Net still have a separate branch for classification but this one does not. 2) This work uses a new loss function, a generalized version of focal loss, to train the network for predicting IoUs. The contribution of each modification to the final performance is unclear. Does the separate branch for classification hurt the performance? What if we just ignore the separate branch for classification in prior works? Is the new loss function essential to the success of the proposed approach? Can the network be trained with other loss such as a simple regression loss (setting the regression target to be zero for negative samples)? Understanding the contribution of each modification would allow us to understand the significance of this work better. Otherwise, the proposed modification seems to be just incremental. The quality of the predicted IoUs is crucial to the performance of the detector. The authors should provide analysis on how well the predicted IoUs match the actual IoUs between the predicted box and the ground-truth. They can do so by calculating the correlations between the predicted and the actual IoUs. Some details on the localization loss function (i.e. DFL) are unclear. What does discretizing the range with even intervals delta (line 166-167) mean? My understanding of DFL is that the network predicts n values and a likelihood for each value. It then calculates the expected value (eq. 5). How does the delta play a role in this equation? Also what does the regression target mean in Fig 5c? How do the authors get the regression target? The second loss function seems to allow the network to fit better to the distribution of the annotations, and hence the network may not generalize well to data from unseen distribution. Having said that, I do not consider this to be a major weakness of this approach, and this is not considered in my rating. I just hope the authors can provide more characteristics of the proposed loss function if possible.

Correctness: Yes. Please see strengths.

Clarity: The part about localization can be improved. The definition of some the terms are unclear which makes it difficult to follow.

Relation to Prior Work: No. Please see weaknesses.

Reproducibility: Yes

Additional Feedback:


Review 2

Summary and Contributions: In this paper, the authors propose to predict IoU instead of pre-assigned class and regress the bbox parameters in a classification manner. The authors also unify the two kinds of loss functions into one form. Generally the idea is novel and the experiments are sufficient. The weakness of this paper is the improvements against several baselines are marginal. But considering there is almost no extra cost, it is still acceptable.

Strengths: Important findings with acceptable solution and sufficient experiements

Weaknesses: I have some concerns about the paper: 1. Why focal loss is used in regression tasks? Focal loss is famous for doing class imbalance problem. It has lower gradients on easy samples, which is a good property for classification. But for regressing the IoU, lower weight for easy samples may cause inaccurate problem. This paper gives me a feeling that the authors only want to have a unified form, but didn't consider the difference between the classification and regression tasks. 2. In [1], the predicted variance of bbox parameters is used for NMS. The algorithm in this paper also produce bbox confidence (sum of two neighbour probabilities). Could it benefit the NMS? 3. The DFL is very similar with softargmax which is widely used in keypoint detection. However, the citation of this research topic is lacking. Please give some credit to authors of keypoint detection papers such as [2] and more. A problem of the softargmax is that the gradient imbalance problem. The form of softargmax is \sum p(x)x. The gradient for p(x) with x=10 is 10 times bigger than the gradient at x=1. This means that the DFL puts more weight on big objects, while the difficult problem of detection is usually the small objects. Overall, the idea of this paper is good but some of the details are still coarse. [1] He Y, Zhang X, Savvides M, et al. Softer-nms: Rethinking bounding box regression for accurate object detection[J]. arXiv preprint arXiv:1809.08545, 2018, 2. [2] Nibali A, He Z, Morgan S, et al. Numerical coordinate regression with convolutional neural networks[J]. arXiv preprint arXiv:1801.07372, 2018.

Correctness: Yes

Clarity: Yes

Relation to Prior Work: Yes

Reproducibility: Yes

Additional Feedback: See above =============================== The rebuttal addresses part of my concerns, but not fully. I will still keep my score borderline towards accept on this paper.


Review 3

Summary and Contributions: This paper works on improving existing one-stage detectors FCOS and ATSS. Two contributions are proposed: 1. merging the centerness head and the classification head with a continuous focal loss; 2. changing the regression representation from float number to 16-bins and integral. Both contributions bring ~0.6mAP improvements on COCO under different settings, and the best performance is improved from 47.7 (ATSS) to 48.2.

Strengths: + The quality focal loss idea is neat. It removed a redundant component from FCOS and ATSS for free with slight performance improvement (+0.3 mAP, 3rd and last rows of Table. 1(a)). + The overall performance (48.2 mAP on COCO) is strong and healthy. + The paper is well written and easy to follow. All figures are clear with comprehensive captions. + The reviewer appreciates that the authors included code in the submission, even though the code contains google drive links that expose one of the author's identity.

Weaknesses: - While the reviewer appreciates the contribution of the quality focal loss, the contribution of distribution focal loss looks far less exciting. It seems just changed the regression representation from a simple float to a complex integral over bins. First of all, this is not new and has been studied in the human pose estimation community (e.g., Integral human pose estimation/ LCRNet). Also, it makes a simple, straightforward representation complex and slower, and has nothing to do with the focal loss idea. The reviewer is not convinced enough to adopt this integral representation into his detector given the minor improvement (+0.3~0.6 mAP) with costs. - Changing from FCOS centerness to IoU brings 0.2~0.4 mAP improvement is interesting. However, this is not highlighted in the paper and the details are hidden in the supplementary. The reviewer found this is a bit misleading in Table. 3, which tends to show the improvement of QFL is 0.7mAP. However 0.4 of them are from changing the FCOS centerness to IoU, and is not the core idea of the proposed continuous focal loss. - Is Figure 3 a real output or just an illustration? The reviewer doubts if the real outputs will be as clean as is shown. - The speed-accuracy trade-off improvement in Figure. 8 and Table. 4 is unclear to the reviewer. E.g., why is X-101-DCN (10 FPS) faster then ATSS (6.9 FPS)? Should it be slower than ATSS or FCOS due to the distribution focal loss? The removal of centerness should be minor in runtime as it is only a single layer, if understood correctly. - It will be beneficial to show the multi-scale testing results as well, to show the proposed method can really push the state-of-the-art number.

Correctness: There are some slight inaccurate claims as mentioned in the paper weakness. The rest of the conclusions are fair as far as the reviewer can access.

Clarity: Yes.

Relation to Prior Work: The reviewer feels the proposed qualitative focal loss based on IoU map has some connections to the modified focal loss used in CornerNet[14] and CenterNet[6]. It will interesting to have a discussion on that.

Reproducibility: Yes

Additional Feedback: Here is how the reviewer justifies the rating: on the one hand, the reviewer likes the continuous focal loss idea and the final performance. However, the "valuable improvement" is only 0.3mAP as discussed in the paper weaknesses. The reviewer feels this is too marginal for a NeurIPS publication. However, the reviewer is happy to raise the score if the authors find considerable misunderstanding in the review. ==================== The rebuttal resolved some of my concerns/ confusions: the run time, Fig. 3, and the multi-scale testing results. However, my complaint on the cumbersomeness of DFL remains, and the overall improvements are still not exciting. Overall, I don't have a strong objection to accepting the paper (I would choose a neutral borderline if these is such option). If the paper is accepted, I highly recommend the authors to make the speed claim clear, and add the analysis of IoU head to the main paper.


Review 4

Summary and Contributions: The paper proposes an approach for object class detection based on existing neural network architectures, but using a novel loss function (generalized focal loss) that combines classification and localization score in a particular way and furthermore represents localization uncertainty explicitly in a non-parametric distribution estimate. Experiments are conducted on the MS COCO detection task and encompass comparison to state of the art results as well as a number of ablations and applications of the proposed loss to existing detection architectures.

Strengths: + The proposed loss function is relatively well motivated and intuitive. It can be applied as an addon to existing detection architectures. + The proposed loss function formulation has a moderate degree of novelty. + The experimental results indicate that the proposed loss function and its ablations are indeed effective and outperform the respective baselines by moderate margins of up to 1.5 percent points in AP. + The experiments contain a plethora of ablations and application of the proposed loss to existing detectors. + The paper makes an effort to illustrate the introduced ideas with diagrams and visualizations.

Weaknesses: - The presentation of the paper could certainly be improved. It contains a multitude of unusual formulations and grammatical oddities (e.g., the very first sentence in the introduction; understanding, however, is not hampered too much by this). - The title is quite misleading: 'qualified and distributed bounding boxes' at least for me set expectations quite differently from what the paper turned out to be about, which is estimating both the quality of and distributions over (bounding box) localization. - The abstract contains too many details of the proposed method and could be shortened.

Correctness: The introduced loss seems plausible on a high level, even though I did not verify every equation explicitly.

Clarity: The paper has some issues in presentation that should be addressed in case a final version were to be prepared.

Relation to Prior Work: The paper gives mostly adequate references to prior work, even though the related work Sect. 2 is on the shorter end of the spectrum.

Reproducibility: Yes

Additional Feedback: I have read the authors' rebuttal, which addresses my concerns concerning the chosen title. I hence decided to stick with my original rating '6: marginally above ...'.

[Author Response · NeurIPS 2020]

Table I: Ablation study about the quality branch and loss type.

| From | Addition Branch | Classification Loss | FCOS [26] | | | ATSS [31] | | |
|---|---|---|---|---|---|---|---|---|
| | | | AP | $AP_{50}$ | $AP_{75}$ | AP | $AP_{50}$ | $AP_{75}$ |
| paper | centerness | FL | 38.5 | 56.8 | 41.6 | 39.2 | 57.4 | 42.2 |
| paper | IoU | FL | 38.7 | 56.7 | **42.0** | 39.6 | 57.6 | **43.0** |
| paper | no | FL | 37.8 | 56.2 | 40.8 | 38.0 | 56.5 | 40.7 |
| rebuttal | no | L2 | 19.2 | 27.4 | 21.0 | 20.4 | 28.6 | 22.1 |
| paper | no | QFL (**ours**) | **39.0** | **57.8** | 41.9 | **39.9** | **58.5** | **43.0** |

Table II: Performance comparisons. ✓ of $MS_{test}$ denotes multi-scale testing.

| From | Method | Backbone | $MS_{test}$ | FPS | AP |
|---|---|---|---|---|---|
| paper | ATSS [31] | X101-32x**8d**-DCN | | 6.9 | 47.7 |
| rebuttal | ATSS [31] | X101-32x**4d**-DCN | | 10.0 | 47.4 |
| paper | GFL (**ours**) | X101-32x**4d**-DCN | | 10.0 | **48.2** |
| rebuttal | ATSS [31] | X101-32x8d-DCN | ✓ | - | 50.7 |
| rebuttal | ATSS [31] | X101-32x4d-DCN | ✓ | - | 50.3 |
| rebuttal | GFL (**ours**) | X101-32x4d-DCN | ✓ | - | **51.1** |

Thanks a lot for the informative and constructive reviews. In general, the reviewers appreciate the novelty and motivation
of GFL, but also raise several concerns about its contribution. We argue that the contributions of GFL are *significant*:
We revisit the common problems in the classification/regression representations of recent dense detectors, and then
provide an effective solution (GFL) for them. The proposed GFL is *simple* (without the need of extra quality estimation
branch), *fast* (cost-free in speed) and *effective* (consistent 0.6∼1.4 AP improvement). It provides a practical and general
format for the detection head, which is compatible for most dense detectors. Therefore, we think that GFL has the
potential to be widely applied in the field of dense object detection.

**To Reviewer #1 Q1**: Ablation study about the separate branch and different forms of loss function. **A1**: We conduct
additional experiments with L2 regression loss to compare against QFL (Table I). Some other related results from Table
1(a) in the original paper are also included in Table I. We observe that (1) ignoring the separate branch considerably
hurts the performance (FCOS: 38.5→37.8, ATSS: 39.2→38.0); (2) the proposed new loss function (QFL) is essential
since the performance of simple L2 regression loss drops dramatically (FCOS: 39.0→19.2, ATSS: 39.9→20.4). The
major reason is that the simple regression loss lacks the good property of Focal Loss and GFL which handle the class
imbalance problem well. We will update these analyses to the paper in the later version.

**Q2**: Quality of the predicted IoUs. **A2**: We calculate the Pearson correlation coefficient between the predicted IoUs and
the actual IoUs (R-50) over all the validation images. The statistic of GFL is 0.78, which is larger than ATSS (0.72). It
shows that GFL indeed improves the quality of the predicted IoUs. We will update these results in the revised version.

**Q3**: Meanings of $\Delta$ and regression target. **A3**: $\Delta = y_{i+1} - y_i, \forall i \in [0, n-1]$. "regression target" is defined in "we
adopt the relative offsets from the location to the four sides of a bounding box as the regression targets", which is in line
156-157 of the original paper. The regression targets are obtained from each positive location and its corresponding gt
bbox, divided by the stride value of its corresponding FPN level. We will make it clearer for readers.

**To Reviewer #2 Q1**: DFL in regression tasks. **A1**: Here the "Focal" in DFL has a completely different meaning: it
forces the network to rapidly *focus* on the probabilities near the target label (line 174-182). Instead, the "Focal" in QFL
means *focusing* the model on hard examples. They have different meanings but share a generalized formulation (GFL).

**Q2**: Could bbox confidence benefit the NMS? **A2**: Yes. We use variance voting method in "Softer-nms" and improve
GFL (R-50) by 0.2 AP. In "Softer-nms", variance voting achieves similar gains for Faster RCNN, i.e., ∼0.3 AP.

**Q3**: Relation between DFL and the keypoint detection paper. **A3**: Although the integral form seems similar in keypoint
detection community, our work is the first to introduce the *integral form* of a *General distribution* into the object
detection field. Meanwhile, we also provide a derivation by extending the concept of Dirac delta distribution, from a
theoretical perspective. Further, we design a novel DFL that quickly focuses on learning probabilities near gt labels. We
will cite and discuss these related works (including LCRNet, etc. mentioned by Reviewer #3) in the revised manuscript.

**To Reviewer #3 Q1**: Unclear speed of methods (ATSS: 6.9 FPS, GFL: 10.0 FPS) with X-101-DCN & multi-scale
testing. **A1**: There are some misunderstanding here. The gap of the speeds only comes from the backbone part of
ATSS (X-101-32x**8d**-DCN) and GFL (X-101-32x**4d**-DCN), where the bottleneck feature dimension of ATSS (32x**8d**)
is **twice** that of GFL (32x**4d**). GFL is cost-free and has the same speed as ATSS under the same backbone (Table 3, II).
Due to GPU memory constraint (11G), we cannot train models with very large backbone (X-101-32x**8d**-DCN). So
we make more comparisons between GFL and ATSS with backbone X-101-32x4d-DCN in Table II. The multi-scale
testing results are also included. We observe that GFL improves ATSS by 0.8 AP under the same X-101-32x4d-DCN
backbone, and the multi-scale testing result of GFL really pushes the state-of-the-art number of ATSS (50.7→51.1).

**Q2**: About Fig. 3. **A2**: Fig. 3 is a real output of the model (R-50). The config and pretrained model are in Supplementary
Material (SM), which can reproduce these figures exactly. More results (Fig. 13, 14) are also provided in SM.

**Q3**: Details of changing from centerness to IoU & limited "valuable improvement". **A3**: The standard version of
FCOS/ATSS defaults to use the centerness branch. Our proposed QFL is designed to optimize a novel "classification-
*quality*" joint representation, whilst identifying the "*quality*" here to be "IoU" instead of "centerness" deserves a
"valuable improvement" because there is no previous work pointing out that issue. Further, we give a deep and statistical
analysis to prove that IoU is better than centerness as a quality measurement (see **Section E** of SM), exposing an
informative fact that centerness has a *fatal flaw* in its definition: centerness can make some quality gt labels too small
(close to 0) to recall a set of objects. For the first time, we provide a convincing reason and suggest that the community
should use IoU instead of centerness although centerness is very successful in FCOS and ATSS. We will add these
analyses into the main paper to make our unique contributions (including the DFL part, see **#2 A3**) clearer.

**To Reviewer #4 Q1**: The presentation. **A1**: Thanks for the kind suggestions about the presentation regarding the title,
abstract and formulations. We will try our best to streamline the abstract and title, and clarify possible formulations.

[Meta-Review · NeurIPS 2020]

Expert reviewers agree that this paper has a novel contribution and lean toward accept. The reviewers like the proposed idea and find the empirical results strong. But the presentation of the paper can be significantly improved. The authors should follow the reviewers' suggestions and significantly revise the paper. In particular, the authors should improve the presentation of the paper following R4's suggestions and add the additional analysis and clarifications as discussed in the rebuttal.